# Multimodal Interventions to Prevent and Control Carbapenem-Resistant Enterobacteriaceae and Extended-Spectrum β-Lactamase Producer-Associated Infections at a Tertiary Care Hospital in Egypt

**DOI:** 10.3390/antibiotics10050509

**Published:** 2021-04-30

**Authors:** Noha A. Kamel, Khaled M. Elsayed, Mohamed F. Awad, Khaled M. Aboshanab, Mervat I. El Borhamy

**Affiliations:** 1Department of Microbiology, Faculty of Pharmacy, Misr International University (MIU), Cairo P.O. Box 19648, Egypt; noha.ahmed@miuegypt.edu.eg (N.A.K.); khaled.elsayed@miuegypt.edu.eg (K.M.E.); mervat.ismail@miuegypt.edu.eg (M.I.E.B.); 2Department of Biology, College of Science, Taif University, P.O. Box 11099, Taif 21944, Saudi Arabia; m.fadl@tu.edu.sa; 3Department of Microbiology and Immunology, Faculty of Pharmacy, Ain Shams University, Organization of African Unity St., Cairo P.O. Box 11566, Egypt; 4International Medical Center, Clinical Microbiology Laboratory, Cairo P.O. Box 11451, Egypt

**Keywords:** CRE, ESBL producers, infection prevention and control measures, tertiary healthcare setting

## Abstract

The current rise of multidrug-resistant (MDR) Gram-negative Enterobacteriaceae including the extended-spectrum β-lactamase (ESBL)-producing organisms and carbapenem-resistant Enterobacteriaceae (CRE) has been increasingly reported worldwide, posing new challenges to health care facilities. Accordingly, we evaluated the impact of multimodal infection control interventions at one of the major tertiary healthcare settings in Egypt for the aim of combating infections by the respective pathogens. During the 6-month pre-intervention period, the incidence rate of CRE and ESBL-producing clinical cultures were 1.3 and 0.8/1000 patient days, respectively. During the post-intervention period, the incidence of CRE and ESBL producers continued to decrease, reaching 0.5 and 0.28/1000 patient days, respectively. The susceptibility rate to carbapenems among ESBL producers ranged from 91.4% (ertapenem) to 98.3% (imipenem), amikacin (93%), gentamicin (56.9%), and tobramycin (46.6%). CRE showed the highest resistance pattern toward all of the tested β-lactams and aminoglycosides, ranging from 87.3% to 94.5%. Both CRE and ESBL producers showed a high susceptibility rate (greater than 85.5%) to colistin and tigecycline. In conclusion, our findings revealed the effectiveness of implementing multidisciplinary approaches in controlling and treating infections elicited by CRE and ESBL producers.

## 1. Introduction

Globally, multidrug-resistant (MDR) organisms are considered to be one of the major public health issues within healthcare settings [1]. Infections with MDR organisms or the rising, extensive, drug-resistant Gram-negative bacteria including the carbapenem-resistant Enterobacteriaceae (CRE) and extended-spectrum β-lactamase (ESBL) producers remain the most challenging. This is because of their limited treatment options, their rapid spread among healthcare environments causing outbreaks associated with high mortality rates and their associated prolonged hospital stays with a high economic burden [2,3,4]. It is also estimated that by 2030, more than half of *Klebsiella pneumoniae* and *Escherichia coli* will be resistant to third-generation cephalosporins [5]. Additionally, the widespread distribution of Enterobacteriaceae in nature and gastrointestinal tracts has facilitated their extensive spread among the ecosystem, causing community and hospital-acquired infections [6,7]. Additionally, prior hospitalization with a prolonged stay and underlying disease conditions makes patients vulnerable to infections [8]. Consequently, preventing the spread of MDR is a major health issue that requires instant efforts to rapidly react against the deadly superbugs. To enable the containment of MDR Gram-negative pathogens (GNP), early detection of CRE—especially for carbapenemase producers and ESBL producers—through targeted patient screening, infection prevention and control measures (IPC) and applying antimicrobial stewardship programs were of proven value for weakening the impact of resistant pathogens among healthcare settings [9,10]. For earlier recognition of the colonization state that usually precedes or co-exists with MDR Gram-negative bacteria, rectal surveillance cultures were examined. Such active surveillance programs play a pivotal role in identifying high-risk patients for developing subsequent infections and shed a light on the importance of earlier implantation of IPC measures [11]. The application of core IPC, including antimicrobial stewardship programs, contact precautions, environmental cleaning and staff education should be applied to minimize the risk of MDR spread between or within health care settings [11,12]. Moreover, MDR carriers, especially CRE, should be accommodated in isolated rooms whenever possible [13]. Therefore, the aim of this study was to evaluate the effect of applying the core IPC measures on the incidence rate of MDR GNP-associated infections including CRE and ESBL producers in one of the major tertiary healthcare settings in Egypt, with the aim of combating these nightmare pathogens.

## 2. Results

### 2.1. Incidence of CRE and ESBL Producers (2017–2019)

During the 6-month pre-intervention period from June 2017 to December 2017, the incidence (median rate) of CRE and ESBL-producing clinical cultures were 1.3 and 0.8/1000 patient days, respectively. During the post-intervention period (January 2018–August 2019), the incidence of CRE and ESBL producers continued to decrease, reaching 0.5 and 0.28/1000 patient days, respectively (Figure 1 and Figure 2).

The rate of distribution of CRE and ESBL producers among the different clinical specimens per 1000 patient days is shown in Figure 3 and Figure 4. The incidence of CRE for a 2-year period post-intervention decreased from 0.27 to 0.13 (blood), 0.39 to 0.26 (sputum), 0.13 to 0.08 (urine) and 0.17 to 0.02 (wound). The incidence of ESBL producers during the 2-year post-intervention period also decreased from 0.17 to 0.08 (blood), 0.11 to 0.06 (sputum), 0.17 to 0.02 (urine) and 0.06 to 0 (wound). Overall, 253 patients were screened for CRE and ESBL colonization within 2 days of their respective International Medical Center (IMC) admissions. Of the 253 patients screened, 58 (23%) and 87 (34%) were colonized with CRE and ESBL producers, respectively. Both CRE and ESBL producers were positive for one or more of CR and ESBL genes as determined by the agarose gel electrophoresis and DNA sequencing of the PCR products. Among the positively screened patients, 21 patients (36%) and 24 patients (27.5%) developed subsequent infection with CRE and ESBL producers, respectively. Out of the 195 and 166 patients negatively screened for CRE and ESBL producers, 22 (11%) and 23 (14%) developed subsequent clinical infections, respectively. Out of the 58 and 87 positively screened CRE and ESBL producers, 33 (57%) and 29 (33.3%) developed subsequent clinical infections, respectively. Out of 195 and 166 negatively screened CRE and ESBL producers, 33 (17%) and 25 (15%) developed clinical infections, respectively.

Out of 66 clinical specimens positive for CRE, about 41% were community-acquired infections (CAIs) and only 9% were hospital-acquired infections (HAIs). Out of 54 clinical specimens positive for ESBLs, 50% were CAIs and about 4% were HAIs.

### 2.2. Antibiogram Analysis of CRE and ESBL Producers 

The most commonly recovered CRE and ESBL producers were *Klebsiella pneumoniae* (80%) and *E. coli* (67.7%). As depicted in Figure 5, the susceptibility rates to carbapenems among ESBL producers ranged from 91.4% (ertapenem) to 98.3% (imipenem). The other tested β-lactam group showed reduced susceptibility that exceeded 86.2%, except for cefoxitin (43.1%) and piperacillin/tazobactam (48.2%). On the other hand, CRE showed the highest resistance pattern toward all of the tested β-lactam groups. Among ESBL producers, the susceptibility rates to amikacin, gentamicin and tobramycin were 93.1%, 56.9% and 46.6%, respectively. In contrast, CRE had recorded a high resistant pattern toward the tested aminoglycosides ranging from 87.3% to 94.5%. Both groups of MDR organisms showed reduced susceptibility toward tested quinolones and sulfamethoxazole/trimethoprim. The overall susceptibility rates to colistin and tigecycline were greater than 85.5% among CRE and ESBL producers. 

The mortality rate within patients infected with ESBL-producing bacteria was 30%, whereas within patients infected with CRE, it was 67%. The highest mortality rates associated with CRE infections were recovered from respiratory tract infections (19, 49%), followed by bloodstream infections (6, 15%), urinary tract infections (8, 21%), surgical site infections (4, 10%), and others including gastrointestinal infections (2, 5%). The highest mortality rates associated with ESBL producers associated infections were recovered from bloodstream infections followed by respiratory tract infections and urinary tract infections.

## 3. Discussion

Like other developing countries, Egypt has limited data regarding the surveillance system that can track the burden of infectious diseases. Moreover, incomplete data about the etiology of Gram-negative, MDR-associated infections, including pneumonia, bloodstream and genitourinary infections could underestimate the problem by at least 50% [14]. Accordingly, this study was undertaken in a tertiary care hospital to examine the effect of implementing care bundles on the improvement of patient health. 

An “all or none approach” was used where each element of the bundle was implemented collectively to achieve the most favorable outcomes [14,15]. A multimodal IPC intervention, such as applying antimicrobial stewardship programs, controlling the source of infection coupled with environmental cleaning and compliance were collectively evaluated. Our study revealed that baseline ESBL producers and CRE rates were 0.8 and 1.3 in the pre-intervention period, and 0.28 and 0.5 in the post-intervention period, respectively. Our results were slightly higher than that reported by a recent study where the ESBL producers and CRE rates were 0.70 and 0.11 in the pre-intervention period and 0.11 and 0.09 in the post-intervention period, respectively [16]. Such discrepancies between these results could be attributed to different sample size and the assessment of diverse interventions that hinders the contribution of infection control measures. Although targeted screening and active surveillance are quite far beyond the capacity of most facilities especially in developing countries, the application of such simple and cost-effective interventions will control the spread of MDR organisms among health care settings as reported by other studies [17,18].

Unlike MDR Gram-positive organisms such as *Staphylococcus aureus* and *Enterococcus* spp., active screening as part of a transmission control program for the MDR Gram-negative pathogens remains controversial. However, the European Society of Clinical Microbiology and Infectious Diseases has strongly recommended that such intervention be followed by contact precautions in epidemic settings [18]. Our results reveal that active rectal screening for CRE and ESBL producers is feasible, with a rate of colonization of 23% and 34%, respectively. Our data were similar to other recent studies, with a rate of CRE colonization reaching 21% [4] and ESBL colonization reaching 30.6% among hospitalized patients [19]. Additionally, our data showed that 36% and 27.5% of the positively screened patients had developed subsequent infection with CRE and ESBL producers, respectively. A higher rate of subsequent CRE infections (47%) was reported by another study, ensuring that rectal colonization could be an important epidemiological risk factor for the development of subsequent CRE infection [20]. Concerning ESBL-producers fecal carriers, one study also found that 27% of patients had acquired infections at an ICU [21]. These findings highlight the contribution of colonizing enterobacterial isolates as one of the potential endogenous reservoirs in the human intestine for subsequent infections. 

Our study revealed that about 41% of the positively screened CRE patients had acquired CRE infections from community settings and 9% from hospital settings. Concerning positively screened cases with ESBL producers, our data revealed that 50% were CAIs and about 4% were HAIs. The level of CAIs with CRE and ESBL producers was slightly higher than other studies that ranged from 12% to 30% [22,23], respectively, and slightly lower than a recent study in India, where 64.5% suffered from community-acquired ESBL producers infections [24]. Our findings assert that MDR pathogens, especially ESBL producers and CRE, have been more recently recognized in community settings even without prior hospital exposure. Hence, clinicians should give attention to the emerging public health problem by identifying predictive risk factors that could improve patient management. Interestingly, a low incidence of HAIs by CRE and ESBL producers was recorded by our study compared to other previous international studies. Such a promising decline in levels could be related to intense infection-control programs that were applied at our tertiary care hospital. 

In this study, the mortality rate within patients infected with CRE was 2.2 times higher than that of ESBLs producing bacteria. Carbapenem-resistant non-bacteremic infections (respiratory tract, urinary tract and wound infections) were associated with 85% mortality and 49% of patients with respiratory tract infections evolved into death. Similar results were reported by Tumbarello et al., as the mortality rate reached 40% among patients with lower respiratory tract infections caused by CR *Klebsiella pneumoniae* [25]. On the other hand, our data showed that the mortality rate among patients infected with ESBL producers that caused bloodstream infections reached 32%. Similar results were also recently reported where the mortality rate of those who suffered from bloodstream infection due to ESBL-producing *E. coli* was 30.8% [26]. However, we should put into consideration that mortality might also be due to the severity of the underlying disease and its co-morbidity. 

In the past few decades, the incidence of MDR Gram-negative associated infections have increased dramatically with international disparity. Thus, regional investigation of antimicrobial resistance patterns coupled with pathogen stratification is of paramount importance to create, estimate and update empirical and exiting treatment guidelines. Overall, our data revealed that the most common pathogens for CRE and ESBL producers cases were *Klebsiella* spp. (80%) and *E. coli* (68%), respectively. Such findings were almost constituent with other findings reporting a high prevalence of *Klebsiella* spp. ranging from 71% [27] to 85% [28] for CRE cases and a high prevalence of *E. coli* reaching 60% among ESBL-producers cases [29]. Regarding the susceptibility profile of ESBL producers producing isolates, our data showed a high resistance pattern for third and fourth generation cephalosporins ranging from 86% to 95%, fluoroquinolones between 74% and 79% and trimethoprim/sulfamethoxazole at 83%. This high pattern of co-resistance could be attributed to a relatively large plasmid that encodes ESBL producers enzymes and genes for other non-β-lactam antibiotics, especially for fluoroquinolones, co-trimoxazole and/or aminoglycosides. However, in accordance with other studies, the majority of ESBL producers showed an enhanced sensitivity pattern toward carbapenems, amikacin, colistin [30,31,32] and tigecycline [33]. In comparison to ESBL producers, CRE antibiogram analysis was more severe and complicated, providing evidence that the latter strains are usually resistant to many other classes of antibiotics in clinical practice [34]. With the exception of colistin and tigecycline, CRE showed a high level of resistance (89% to 100%) to other tested antimicrobial classes. Like our findings, other studies have also reported that tigecycline and colistin were the most effective antimicrobial agents against CRE [34,35]; however, literature has reported that tigecycline alone was associated with a high mortality rate and decreased clinical efficacy [36], while renal impairment and the appearance of hetero-resistance were the main drawbacks of colistin monotherapy [37]. In support of this data, clinicians should examine the use of such mainstay antimicrobials agents in combinations to lessen the emergence of high-level resistant phenotypes.

## 4. Materials and Methods

### 4.1. Study Design

A prospective study was conducted at the International Medical Center (IMC) from July 2017 to June 2019 to screen carriers or patients infected with ESBL-producing Gram-negative pathogens and CRE. The IMC is an acute care, tertiary hospital with around 800 beds, 10 different ICUs and 3 different wards (oncology, surgical and medical). Authorized members of an infection control team were assigned to collect the microbiological records including the identified culture with its sensitivity pattern from different clinical specimens, including blood, sputum, urine and wound swabs among all ICUs and wards of IMC. The study protocol was reviewed and approved by the institutional ethics committee, Faculty of Pharmacy, Ain Shams University (ENREC-ASU-2018-72). This study was conducted in accordance with the ethical principles stated in the Declaration of Helsinki. 

### 4.2. Microbiological Procedures and Definitions

Rectal swab specimens were collected within 48 hours of ICU admission to assess bacterial colonization from 253 studied participants over a period of 6 months, from July 2017 to December 2017. Swabs were inserted 1 cm into the rectum and then rotated 360°. Amies transport medium was used for immediate transformation for further laboratory analysis. For the rapid detection of CRE and ESBL producers from potentially colonized rectal specimens, the HB&L carbapenemase and ESBL producers-AmpC kits (Alifax, Padua, Italy) were used, respectively. Kits were read by a ALIFAX microbiology line analyzer (Alifax, Padua, Italy) that depends on laser-scattering technology for the automatic detection and enumeration of tested broth vials. Briefly, the laser beam passes through a glass bottle containing the reagent (HB&L carbapenamase or ESBL producers-AmpC) in addition to the specimen and a selective supplement broth that contain certain antibacterial and antifungals agents. For CRE and ESBL detection, the broth was combined with carbapenem and certain 3rd-generation cephalosporins, respectively, to inhibit non-carbapenem resistant Enterobacteriaceae spp. and non-ESBL. The patented technology enabled us to monitor real-time growth curves and positive or negative results were reported based on a calculation of colony-forming unit per mL. As recommended by the manufacturer, we have used the automated system, Vitek-2 (bioMérieux, Marcy L’Etoile; Lyon, France) for the identification and detection of antimicrobial breakpoints. 

By definition, the tested strains were considered MDR if they displayed resistance to 3 or more antimicrobial classes, i.e., susceptible to not more than 1 of the following classes by disk diffusion method: aminoglycosides (10 µg), carbapenems (10 µg), cephalosporins (30 µg), fluoroquinolones (5 µg) or penicillin (10 µg) [38]. Susceptibility to polymyxins, sulbactam, sulfonamides and tetracycline were not considered in the definition. The rate of MDR organisms was calculated monthly by dividing the number of MDR by the total number of patient days and multiplying by 1000 [39]. Admitted persons were classified into colonized and cases with infections according to previously published literature [40]. Infections with CRE and ESBL producers were defined as the detection of Enterobacteriaceae isolates resistant to carbapenem antibiotics and 3rd-generation cephalosporins, respectively, with additional signs and symptoms of infection as determined by the clinicians. Detected infection from days 1–3 upon IMC admission was counted as community-acquired (CAIs), while later detections counted as hospital-acquired (HAIs) [41]. The recovered bacterial isolates were tested for susceptibility to the antimicrobial agents recommended by the Clinical and Laboratory Standards Institute (CLSI, 2018) for each bacterial species [42].

### 4.3. Molecular Identification of ESBL and CR Genes

#### 4.3.1. Extraction of Chromosomal DNA from CRE and ESBL Producers

Overnight cultures of the phenotypically confirmed CRE or ESBL-producing isolates were grown in Luria Bertani broth containing 16 μg/mL meropenem or ampicillin, respectively. The chromosomal DNA was carried out using a GeneJET DNA extraction kit (Thermo Fisher Scientific, Waltham, MA, USA). The extracted chromosomal DNA was analyzed via agarose gel electrophoresis and visualized by a UV transilluminator [43].

#### 4.3.2. PCR Detection of ESBL and CR Genes

The extracted chromosomal DNA of each isolate was used as a template for PCR using the appropriate primers synthesized by Invitrogen (Thermo Fisher Scientific, Waltham, MA, USA), and Dream *Taq* Green PCR Master Mix (Thermo Fisher Scientific, Waltham, MA, USA). The annealing temperatures (Ta) and primers for ESBL genes—including *bla*_CTX-m_*, bla*_TEM_ and *bla*_SHV_ as well as for 5 of the major CR genes, including *bla*_KPC_, *bla*_NDM_, *bla*_VIM_, *bla*_OXA-48_ and *bla*_IMP_—are shown in Table 1. The amplified PCR products were analyzed using agarose gel electrophoresis and the size of DNA fragments was determined by comparing them to a 100 bp DNA ladder (GeneRuler 100 bp DNA ladder, Thermo Fisher Scientific, Waltham, MA, USA).

#### 4.3.3. Sequencing of Selected PCR Products

Some PCR products of amplified genes were sent for sequencing at Macrogen Inc. (Seoul, Korea) using an Applied Biosystems 3730XL sequencer. The assembly of the obtained forward and reverse sequence files into the final consensus sequence was completed using BioEdit v7.2.5 software [50]. The open reading frames (ORFs) of the final contigs were detected using FramePlot 2.3.2 [51]. The sequencing data were analyzed using the basic local alignment search tool.

### 4.4. Infection Control and Preventitive Interventions

A multidisciplinary team approach, including an infection control department, microbiology laboratory and clinical pharmacy was necessary to develop and implement strategies to control the spread of CRE and ESBL producers among the hospital settings. A pilot study was undertaken for 6 months from July 2017 to December 2017 and a prospective period from January 2018 to August 2019. The following core IPC measures were implemented during the pilot study and enhanced during the second part of the study.

#### 4.4.1. Contact Isolation

The infection control team had trained, monitored and provided feedback on hand hygiene and the use of gloves and gowns [52,53]. In case of carriage or infection with CRE, patient and staff cohorting were applied and such patients were preferably placed in a single room [54,55]. 

#### 4.4.2. Environmental Cleaning

The infection control team checked that beds and the surroundings were disinfected by chloride detergents. Cultures from different surfaces were randomly taken 2 times during the pilot study [56]. On a daily basis, the new checklist containing all items in the devices bundle was audited by the infection-control team.

#### 4.4.3. Chlorhexidine Bathing

On a routine daily basis, 2% chlorhexidine gluconate (impregnated cloth) was used for whole-body bathing by certified nursing assistants [57,58].

#### 4.4.4. Antimicrobial Stewardship Programs

The clinical pharmacy department started to update the list of restricted antimicrobials and the checklist of compliance. Restricted broad-spectrum antibiotics including 3rd or 4th-generation cephalosporins, carbapenems and fluoroquinolones were only prescribed after approval by a designated infectious diseases physician. Additionally, the clinical department communicated with the microbiology laboratory to update the list of antibiotic combinations and a new pocket pamphlet was prepared to promote the antimicrobial stewardship program. 

#### 4.4.5. Statistical Analyses

Data analyses were performed using GraphPad Prism (version 6). Qualitative variables were expressed as frequency and percentages. The incidence density was calculated as infections per 1000 patient days (stratified by infection type).

## 5. Conclusions

Being a carrier or infected with MDR Enterobacterales, particularly ESBL producers and CRE, requires prompt action by health care settings to control transmission and optimize patient management protocols. The establishment of a multidisciplinary approach, including screening, contact isolation, environmental cleaning and applying antimicrobial stewardship programs showed promising results in reducing the incidence of CRE and ESBL producers per 1000 patient days. Our findings revealed the effectiveness of multi-modal IPC approaches (controlling the source of infection, environmental cleaning and applying antimicrobial stewardship programs) in controlling and preventing infections elicited by CRE and ESBL producers, and hence they will be of great value for guiding clinicians in implementing effective treatment options against the respective clinically relevant pathogens.

## Figures and Tables

**Figure 1 antibiotics-10-00509-f001:**
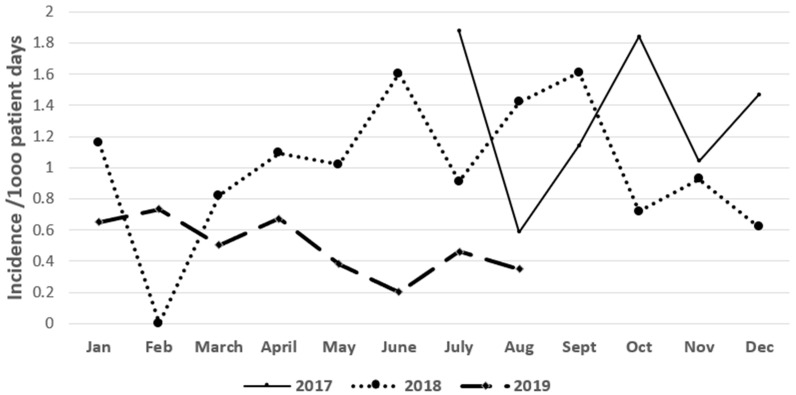
Incidence rate of CRE from the period 2017–2019.

**Figure 2 antibiotics-10-00509-f002:**
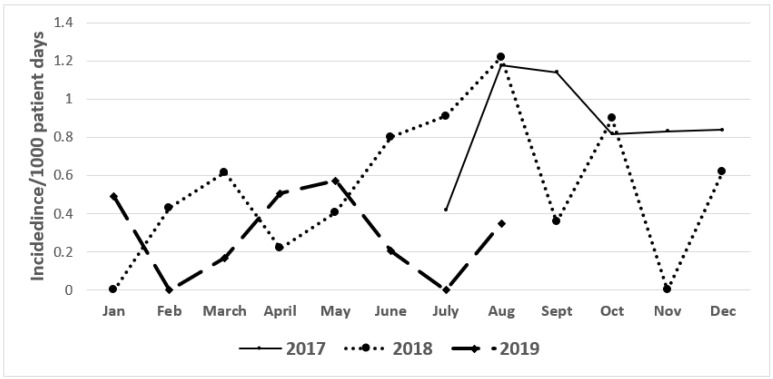
Incidence rate of ESBL producers from the period 2017–2019.

**Figure 3 antibiotics-10-00509-f003:**
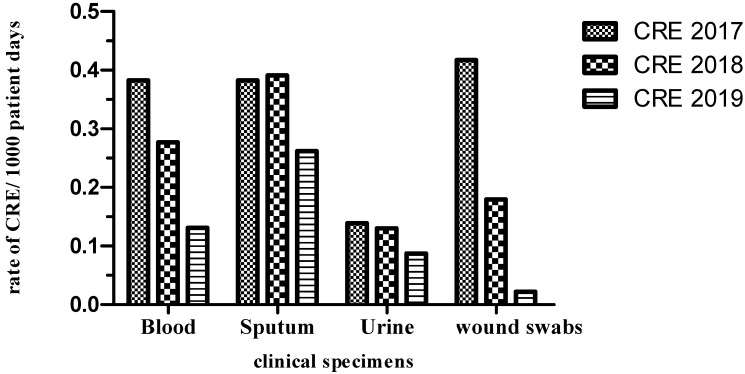
Rate of CRE distribution among various clinical specimens per 1000 patient days over the period 2017–2019.

**Figure 4 antibiotics-10-00509-f004:**
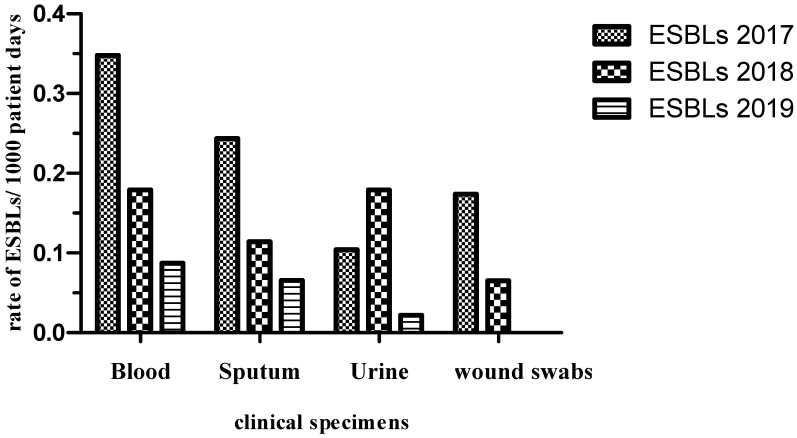
Rate of ESBL producers distribution among various clinical specimens per 1000 patient days over the period 2017–2019.

**Figure 5 antibiotics-10-00509-f005:**
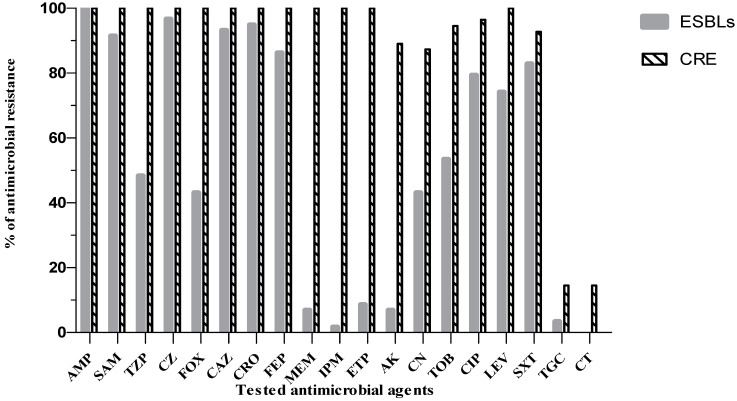
Antibiogram analysis of CRE and ESBL producers against various antimicrobial agents. AMP: Ampicillin, SAM: Ampicillin/Sulbactam, TZP: Piperacillin/Tazobactam, CZ: Cefazolin, FOX: Cefoxitin, CAZ: Ceftazidime, CRO: Ceftriaxone, FEP: Cefepime, MEM: Meropenem, IPM: Imipenem, ETP: Ertapenem, AK: Amikacin. CN: Gentamicin, TOB: Tobramycin, CIP: Ciprofloxacin, LEV: Levofloxacin, SXT: Trimethoprim/Sulfamethoxazole, TGC: Tigecycline, CT: Colistin.

**Table 1 antibiotics-10-00509-t001:** Primers, expected PCR product sizes and annealing temperatures (T_a_) of the tested genes.

Gene	Primer	Primer Sequence (5’ → 3’)	Expected PCR Product Size (bp)	T_a_ (°C)	References
*bla* _KPC_	P_f_	TGTCACTGTATCGCCGTC	1011	50	[44]
	P_r_	CTCAGTGCTCTACAGAAAACC			
*bla* _NDM_	P_f_	GGTTTGGCGATCTGGTTTTC	621		[45]
	P_r_	CGGAATGGCTCATCACGAT			
*bla* _VIM_	P_f_	TCTACATGACCGCGTCTGTC	748	50	[46]
	P_r_	TGTGCTTTGACAACGTTCGC			
*bla* _OXA-48_	P_f_	GCGTGGTTAAGGATGAACAC	438		[44]
	P_r_	CATCAAGTTCAACCCAACCG			
*bla* _IMP_	P_f_	CTACCGCAGCAGAGTCTTTG	587	50	[47]
	P_r_	AACCAGTTTTGCCTTACCAT			
*bla* _CTX-m_	P_f_	CGCTTTGCGATGTGCAG	550	47	[48]
	P_r_	ACCGCGATATCGTTGGT			
*bla* _TEM_	P_f_	ATGAGTATTCAACATTTCCG	867	47	[49]
	P_r_	CTGACAGTTACCAATGCTTA			
*bla* _SHV_	P_f_	GGTTATGCGTTATATTCGCC	867	47	[49]
	P_r_	TTAGCGTTGCCAGTGCTC			

Notes: *bla*_KPC_, *bla*_NDM_, *bla*_VIM_, *bla*_OXA-48_ and *bla*_IMP_: gene codes for KPC, NDM, VIM, OXA-48-like, and IMP carbapenemases, respectively. *bla*_CTX-m_: gene coding for cefotaxime (CTX-M) extended-spectrum β-lactamase, *bla*_TEM_: gene coding for TEM extended-spectrum β-lactamase, *bla*_SHV_: gene coding for SHV extended-spectrum β-lactamase, P_f_: forward primer, P_r_: reverse primer, T_a_: annealing temperature.

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
