# Peer review of "Multimodal Interventions to Prevent and Control Carbapenem-Resistant Enterobacteriaceae and Extended-Spectrum β-Lactamase Producer-Associated Infections at a Tertiary Care Hospital in Egypt"

_antibiotics, 2021, doi:10.3390/antibiotics10050509_

Round 1

Reviewer 1 Report

The manuscript “Evaluation of multidisciplinary interventions to minimize carbapenem resistant Enterobacteriaceae and Extended spectrum β-lactamase producer-associated infections at a tertiary care hospital in Egypt” focuses on an interesting and important topic. However, there are many shortcuts in the text which need the correction. Moreover, some lacks in the Methods section needs additional explanation and supplementation. The detailed comments have been added to separate file.

Major comments:

  1. Please add some details to Methods section about identification of ESBL and CRE. Did you only analyse based on the phenotypic method without any molecular confirmation? If yes, your results are only preliminary studies and need to be confirmed by using molecular methods.

  1. Please add the results of statistical analyses.

Author Response

Authors’ Response to Reviewer 1 Comments

Dear Dr. Icey Li, Assistant Editor, MDPI, Antibiotics,

On the behalf of all authors, we would like to thank the reviewers for their valuable comments and suggestions that improve and add to this manuscript. Corrections are track-changed, highlighted in yellow color for reviewer 1. All corrections have been included in the revised manuscript.

Reviewer 1 comment

  • The manuscript “Evaluation of multidisciplinary interventions to minimize carbapenem resistant Enterobacteriaceae and Extended spectrum β-lactamase producer-associated infections at a tertiary care hospital in Egypt” focuses on an interesting and important topic. However, there are many shortcuts in the text which need the correction. Moreover, some lacks in the Methods section needs additional explanation and supplementation. The detailed comments have been added to separate file.

Authors’ response:

Thanks for your valuable comment. The whole manuscript had been revised thoroughly to correct shortcuts in text, give detailed information about method as requested by reviewer. Additionally, your valuable comments had been corrected throughout the whole manuscript as follows.

  1. Reviewer detailed comments of separate file.
  2. Among 66 cases with CRE isolates, half cases were positively screened. What does it mean that cases were positively screened, please remove mental shortcuts from the text of your manuscript?

Author response:

Thanks for. The number of specimens were inserted, and the text was corrected as follows:

Out of 66 clinical specimens positive for CRE, about 41% were community-acquired infections (CAIs) and only 9% were hospital-acquired infections (HAIs).  Out of 54 clinical specimens positive for ESBLs, 50% were CAIs and about 4% were HAIs. (page 3, lines, 91-93)

  1. MDRO please add explanation of the abbreviation when you use it first time in the text

Authors’ response

It was corrected in the whole manuscript to from MDRO to MDR organisms (all changes are highlighted in yellow color)

  1. Moreover, incomplete data about etiology of Gram negative MDR associated infections including pneumonia, blood stream and genitourinary infections could underestimate the problem by at least 50%. please add references as a source of this information

Authors’ response:

A reference (Talaat et al, 2016) has been added: (page 6, line 133) and in the reference section (reference number 14)

  1. A prospective study was conducted among ESBL producers

please explain what do you mean using the phrase "ESBL producer"? In standard only microorganisms are the ESBL producer.

Authors’ response:

Thanks for informing us with this valuable information and we had corrected the sentence as follows: A prospective study was conducted at International Medical Center (IMC) from July 2017 to June 2019 to screen carriers or patients infected with ESBL producing Gram negative pathogens and CRE. (page 7, lines, 221-223)

  1. Rectal swab specimens were collected within 48 hours of ICU admission to assess bacterial colonization from 253 studied participants. Swabs were inserted 1 cm into the rectum and then, rotated 360°. please divide the number of samples according to time of collecting samples

Authors’ response:

The sentence was modified to include the number of samples and time period of their collections as follow: “Rectal swab specimens were collected within 48 hours of ICU admission to assess bacterial colonization from 253 studied participants over a period of six months, from July 2017 till December 2017”. (inserted in the revised version, page 8, lines 232-234)

  1. please add in brackets the level of concentration of each antibiotic used in the analyses

Authors’ response:

Backets have been inserted for the concentration of each antibiotic (page 8, lines, 250-253). (also highlighted in yellow)

Major comments:

 Please add some details to Methods section about identification of ESBL and CRE. Did you only analyse based on the phenotypic method without any molecular confirmation? If yes, your results are only preliminary studies and need to be confirmed by using molecular methods.

Authors’ response

Thank you for the reviewer comments. The ESBL and CRE were identified phenotypically and confirmed by molecular methods. More information was given and inserted in the revised manuscript for the phenotypic kits which were read by ALIFAX microbiology line analyzer (Alifax, Padua, Italy) that depends on laser scattering technology for automatic detection and enumeration of tested broth vials (page 8, lines, 246-257). Also, more details for the molecular identification of ESBL and CR genes has been added to the methods section (page 8 and 9; lines 275-291).

  1. Please add the results of statistical analyses.

Statistical analysis of the results is inserted in the result section as requested.

Dear Dr. Icey Li, Assistant Editor, MDPI, Antibiotics,

On behalf of all coauthors who participated in writing this manuscript, we would like to express our deepest gratitude to the reviewer valuable comments that enriched the revising process. All comments and recommendations were done and highlighted in the revised manuscript R1.

Sincerely,

Prof. Dr Khaled Aboshanab (PhD)

Professor of Microbiology and Immunology, Vice Dean for postgraduate Studies and Scientific Research, Faculty of Pharmacy, Ain Shams University, Cairo, Egypt

Reviewer 2 Report

The paper presents an interesting study into infections at a tertiary care hospital in Egypt. The abstract and introduction provide an appropriate level of detail to understand the issue and how this will be a problem in the future.

The results presented are appropriate and well described. There are some issues with one or two of the graphs which would benefit from extra information which should be included for clarity. The discussion also provides an appropriate summation and outline of the findings from the work which has been undertaken.

However, the biggest issue is that the paper outlines the problem and the results which are obtained from the multidisciplinary interventions, but, there is no discussion about these, which one(s) were the most effective?? Why might one form provide better results than the others?? These questions along with a full discussion about this aspect of the paper need to be included as currently, the title is very misleading.

At this point, I feel the paper needs major revision and the inclusion of extra information described above before it could be published, especially with the current title.

Author Response

Authors’ Response to Reviewer 2 Comments

Dear Dr. Icey Li, Assistant Editor, MDPI, Antibiotics,

On the behalf of all authors, we would like to thank the reviewers for their valuable comments and suggestions that will improve and add to this manuscript. Corrections are track-changed, highlighted in green color for reviewer 2.  All corrections have been included in the revised manuscript.

Reviewer 2 comments:

The paper presents an interesting study into infections at a tertiary care hospital in Egypt. The abstract and introduction provide an appropriate level of detail to understand the issue and how this will be a problem in the future. The results presented are appropriate and well described. There are some issues with one or two of the graphs which would benefit from extra information which should be included for clarity. The discussion also provides an appropriate summation and outline of the findings from the work which has been undertaken.

Authors’ response

We would like to thank the reviewer for the valuable and positive comments that will add to the work. The whole manuscript was revised, and all comments were corrected as follows:

  1. Reviewer detailed comments of separate file
  2. Setting (line 18):

Authors’ response

Corrected to settings (page 1, lin2, 18)

  1. Organism (line 33)

Authors’ response: corrected to organisms. (page 1, line 34)

  1. considered to be one of the most opposing events among (line 33) corrected to:

            Authors’ response :Corrected to:

are considered to be one of the major public health issues within healthcare settings . (page 1, lines 34 and 35)

  1. Remained (line 36)

Authors’ response

corrected to remain

  1. Additionally, the ubiquitous of Enterobacteriaceae in nature and gastrointestinal tract had facilitated their widespread among ecosystem, causing (line 41)

Authors’ response

Corrected to: Additionally, the widespread of Enterobacteriaceae in nature and gastrointestinal tract had facilitated their extensive spread among the ecosystem, causing……(page 1, lines 42 and 43)

  1. The rectal surveillance cultures were used for early recognition of colonization state that usually precedes or co-exist with MDR Gram negative bacteria. Additionally, identification of the colonized patients is a helpful tool to determine high risk patients for subsequent infections that needs earlier introduction of IPC measures. (line 51)

Authors’ response

The paragraph has been rephrased as requested (highlighted below) and a reference was added in the revised manuscript..

For earlier recognition of colonization state that usually precedes or co-exist with MDR Gram negative bacteria, rectal surveillance cultures were examined. Such active surveillance programs plays a pivotal role to identify high risk patients for developing subsequent infections and shed light on earlier implantation of IPC measures. (page 2, line 52-55). Reference Tacconelli et al 2014 was inserted (reference number 11 in the reference section).

  1. for 2 years period

Authors’ response

Corrected to: for a 2 year period (page 3, line 77)

  1. Overall, 253 patients were screened for CRE and ESBL producers carriage …(line 79

Authors’ response

Corrected to: Overall, 253 patients were screened for CRE and ESBL colonization (page 3, line 82)

  1. figure 5 (line 100)

Authors’ response

Corrected to Figure 5 (page 4, line 106)

  1. Abbreviation of antibiotics

Authors’ response

All antibiotics abbreviations have been inserted in the legend of Figure 4 as requested (page 5, lines, 118-121)

  1. MDRO (line 142)

Authors’ response

Changed to: MDR organisms in the whole manuscript. (all changes were highlighted).

  1. Enterococcus (line 144)

Authors’ response

Changed to: Enterococcus (page 6, line 157)

  1. Had (147)

Authors’ response

Changed to : has (page 6, line 160)

  1. CAIs, HAIs

Authors’ response

The 2 terms have been previously described and written in full sentences. (page 3, lines 94 and 95)

  1. Our findings revealed the effectiveness of multi-disciplinary approaches in controlling and treatment the infections elicited by CRE (line 288)

Authors’ response

Corrected to: Our findings revealed the effectiveness of multi-modal IPC approaches (controlling source of infection, environmental cleaning, applying antimicrobial stewardship programs) in controlling and treating infections elicited by CRE and ESBL producers and hence….(page 10, lines 350-353).

  1. However, the biggest issue is that the paper outlines the problem and the results which are obtained from the multidisciplinary interventions, but, there is no discussion about these, which one(s) were the most effective?? Why might one form provide better results than the others?? These questions along with a full discussion about this aspect of the paper need to be included as currently, the title is very misleading.At this point, I feel the paper needs major revision and the inclusion of extra information described above before it could be published, especially with the current title.

Authors’ response

Thanks for the reviewer’s suggestion. The title was corrected as “Multimodal interventions to prevent and control carbapenem resistant Enterobacteriaceae and Extended spectrum β-lactamase producer-associated infections at a tertiary care hospital in Egyptfor more clarification and reflection of the work done. In addition. By reviewing previous studies and available literature, we found that for better outcome the IPC program can be implemented multimodally where an integrated approaches or interventions with several components are evaluated collectively. (page 1, line 2)

Dear Dr. Icey Li, Assistant Editor, MDPI, Antibiotics,

On behalf of all coauthors who participated in writing this manuscript, we would like to express our deepest gratitude to the reviewer valuable comments that enriched the revising process. All comments and recommendations were done and highlighted in the revised manuscript R1.

Sincerely,

Prof. Dr Khaled Aboshanab (PhD)

Professor of Microbiology and Immunology, Vice Dean for postgraduate Studies and Scientific Research, Faculty of Pharmacy, Ain Shams University, Cairo, Egypt

Round 2

Reviewer 1 Report

The manuscript was corrected according to all my comments, therefore, I suggest to accept the manuscript at the present form.

Reviewer 2 Report

Having reviewed the changes put forward by the authors I feel that the paper now better reflects the data presented and consequently I would recommend publication following a final review of the English grammar and spelling as some errors still exist.